# Treadmill Running in Established Phase Arthritis Inhibits Joint Destruction in Rat Rheumatoid Arthritis Models

**DOI:** 10.3390/ijms20205100

**Published:** 2019-10-15

**Authors:** Yuta Fujii, Hiroaki Inoue, Yuji Arai, Seiji Shimomura, Shuji Nakagawa, Tsunao Kishida, Shinji Tsuchida, Yoichiro Kamada, Kenta Kaihara, Toshiharu Shirai, Ryu Terauchi, Shogo Toyama, Kazuya Ikoma, Osam Mazda, Yasuo Mikami

**Affiliations:** 1Department of Orthopaedics, Graduate School of Medical Science, Kyoto Prefectural University of Medicine, Kawaramachi-Hirokoji, Kamigyo-ku, Kyoto 602-8566, Japan; y-fujii@koto.kpu-m.ac.jp (Y.F.); hinoue@koto.kpu-m.ac.jp (H.I.); s-shimo@koto.kpu-m.ac.jp (S.S.); tuchi-kf@koto.kpu-m.ac.jp (S.T.); kamada-y@koto.kpu-m.ac.jp (Y.K.); kaihara5@koto.kpu-m.ac.jp (K.K.); shirai.t77@gmail.com (T.S.); ryutel@mbox.kyoto-inet.or.jp (R.T.); shogot@koto.kpu-m.ac.jp (S.T.); kazuya@koto.kpu-m.ac.jp (K.I.); 2Department of Sports and Para-Sports Medicine, Graduate School of Medical Science, Kyoto Prefectural University of Medicine, Kawaramachi-Hirokoji, Kamigyo-ku, Kyoto 602-8566, Japan; shushi@koto.kpu-m.ac.jp; 3Department of Immunology, Graduate School of Medical Science, Kyoto Prefectural University of Medicine, Kawaramachi-Hirokoji, Kamigyo-ku, Kyoto 602-8566, Japan; tsunao@koto.kpu-m.ac.jp (T.K.); mazda@koto.kpu-m.ac.jp (O.M.); 4Department of Rehabilitation Medicine, Graduate School of Medical Science, Kyoto Prefectural University of Medicine, Kawaramachi-Hirokoji, Kamigyo-ku, Kyoto 602-8566, Japan; mikami@koto.kpu-m.ac.jp

**Keywords:** rheumatoid arthritis, exercise therapy, autoimmune disorder, treadmill running, exercise, articular cartilage, collagen-induced arthritis, pro-inflammatory cytokine, connexin 43, osteoporosis

## Abstract

Exercise therapy inhibits joint destruction by suppressing pro-inflammatory cytokines. The efficacy of pharmacotherapy for rheumatoid arthritis differs depending on the phase of the disease, but that of exercise therapy for each phase is unknown. We assessed the differences in the efficacy of treadmill running on rheumatoid arthritis at various phases, using rat rheumatoid arthritis models. Rats with collagen-induced arthritis were used as rheumatoid arthritis models, and the phase after immunization was divided as pre-arthritis and established phases. Histologically, the groups with forced treadmill running in the established phase had significantly inhibited joint destruction compared with the other groups. The group with forced treadmill running in only the established phase had significantly better bone morphometry and reduced expression of connexin 43 and tumor necrosis factor α in the synovial membranes compared with the no treadmill group. Furthermore, few cells were positive for cathepsin K immunostaining in the groups with forced treadmill running in the established phase. Our results suggest that the efficacy of exercise therapy may differ depending on rheumatoid arthritis disease activity. Active exercise during phases of decreased disease activity may effectively inhibit arthritis and joint destruction.

## 1. Introduction

Rheumatoid arthritis (RA) is an autoimmune disorder that particularly affects the synovial membranes. Affected synovial membranes produce an excess of pro-inflammatory cytokines and chemokines, such as tumor necrosis factor (TNF)-α, interleukin (IL)-6, IL-1b, and stromal cell-derived factor 1, which destroy the joints and cause pain and a restricted range of motion [1,2,3]. Other than the joints, RA also affects major internal organs, including the heart, lungs, liver, and brain [4]. The progress of these arthritic and systemic symptoms reduces patients’ activities of daily living (ADL), and because it causes a variety of disabilities, RA is a disease of public health importance, and it must be managed as such.

In RA treatment, priority is given to pharmacotherapy. In recent years, a paradigm shift has occurred in RA pharmacotherapy due to the arrival of biological disease-modifying anti-rheumatic drugs (bDMARDs), and it has become possible to minimize joint destruction [5,6,7]. However, bDMARDs have the disadvantages of adverse drug reactions and high costs, and they can only be used in limited cases [8,9]. Exploring other modalities of treatment of RA is therefore becoming increasingly important. One of such methods is exercise therapy. Exercise therapy is a safe and economical treatment that is widely used for the treatment of several systemic diseases; not only joint diseases. Exercise therapy was conducted based on the experience of individual doctors or therapists. In recent years, its mechanisms are becoming clearer, and evidence is accumulating. Among the various available exercise options, exercise therapy using treadmills works to protect articular cartilage and subchondral bone [10]. In osteoarthritis (OA), it inhibits the formation of bone spurs and bone destruction [11], and it thereby reduces joint symptoms. Treadmill exercise has also been reported to improve cardio-pulmonary function and ADL for patients with heart or lung disease [12,13]. Exercise therapy for RA is strongly recommended by a Cochrane review [14], and it has been reported to improve bodily functions and ADL, clinical indicators of joint symptoms, muscle strength, and cardio-pulmonary function [15,16]. In recent years, treadmill running in rat RA models was reported to inhibit the production of connexin 43 (Cx43) and TNF-α in the synovial membranes, as well as prevent the degeneration of articular cartilage and subchondral bone [17]. It was also revealed that exercise therapy inhibits joint destruction through bio-molecular mechanisms by suppressing pro-inflammatory cytokines.

The natural course of RA is divided into an induction phase following immunological sensitization, a pre-arthritis phase during which autoantibodies are produced, and an established phase during which arthritis occurs and progresses [18]. The effectiveness of treatment may vary with the phase of disease. Dekkers et al. used animal models to study the differences in efficacy of pharmacotherapy for each phase, and they established that pharmacotherapy is more effective in the established phase than in the pre-arthritis phase [19]. It was found that compared with the pre-arthritis phase, in which cytokine production increases sharply, it is easier to get efficacy from anti-TNF-α antibodies in the established phase, during which cytokine quantities are stable. Thus, phase-related differences in cytokine production influence the efficacy of pharmacotherapy [20]. However, phase-related differences in the efficacy of exercise therapy are unknown. Since exercise therapy also demonstrates efficacy in suppressing cytokines, we posed the hypothesis that it would be more effective to suppress arthritis and joint destruction with treadmill running during the established phase than in the pre-arthritis phase.

Based on the above context, the objective of this study was to assess the differences in efficacy of treadmill running during each phase of RA, using rat RA models.

## 2. Results

### 2.1. Kinetic Change in Body Weight and Paw Volume

Body weight in each group (no intervention (control) group, pre-arthritis intervention short (PAS) group, pre-arthritis intervention long (PAL) group, and therapeutic intervention (T) group) gradually increased from day 0 to day 12, decreased from day 12 to day 20, and increased again starting on day 21. Paw volume in each group increased from day 14, reaching its maximum between day 20 and day 23, and gradually decreasing thereafter. No significant differences in either body weight or paw volume were observed among groups from day 0 to day 42 (Figure 1A–C).

### 2.2. Effect of Treadmill Running on Articular Cartilage

To assess the histological effects of treadmill running on joints and their cartilage, we stained the articular cartilage from the rat paws with hematoxylin and eosin (H&E), and safranin O on day 42 (Figure 2A–D). In the control group and the PAS group, we observed intra-articular infiltration by inflammatory cells and pannus formation in hyperplastic synovial membranes. Joint structure destruction was also more severe in these groups than in the T group and the PAL group. Furthermore, safranin O staining in the control and PAS groups was less than in the T and PAL groups. The histological scores were significantly lower in the T and PAL groups than in the control and PAS groups. There was no difference in histological score between the PAS and control groups (Figure 2E,F).

### 2.3. Influence of Treadmill Running on the Production of TNF-α and Cx43 in the Synovium

Following past reports, we conducted immunostaining for TNF-α and Cx43 [17]. TNF-α expression was aggravated throughout the synovial membranes, but the level of staining was significantly lower for the T group than for the control group (Figure 3A,C). For Cx43, the level of staining was significantly lower for the T group than for the control and PAS groups (Figure 3B,D).

### 2.4. Prevention of Bone Loss by Treadmill Running

We used micro-computed tomography (μ-CT) to assess periarticular skeletal composition changes (Figure 4). The T group had higher bone volume fraction (BV/TV), trabecular thickness (Tb.Th), and bone mineral content per tissue volume (BMC/TV) than the control group (Figure 4A,B,D). Furthermore, marrow star volume (MSV) values were significantly lower for the T group than for the control group (Figure 4C).

### 2.5. Effects of Treadmill Running on Bone Erosion

We used μ-CT to assess the effects of treadmill running on bone erosion (Figure 5). The eroded bone surface per repaired bone surface (Es/Rps) value for T group was significantly lower than that of the control group (Figure 5B,C). To assess its effects on bone metabolism, we conducted immunostaining with cathepsin K, one of the osteoclast marker that serves as an indicator of the degree of bone resorption [21,22]. Many cathepsin K-positive cells were observed in the pannus areas of the control and PAS groups. Fewer cathepsin K-positive cells were in the T and PAL groups than in the control and PAS groups (Figure 5A).

## 3. Discussion

In this study, we conducted comparative analyses on four groups of rat RA models to assess the differences in the efficacy of treadmill running treatment in each phase of RA. We studied a control group that was raised freely after immunological sensitization, a PAS group that did treadmill running only during the pre-arthritis phase, a PAL group that did treadmill running from the pre-arthritis phase to the established phase, and a T group that did treadmill running only during the established phase. The results revealed that there were no differences in body weight among the groups, and that even if 12 m/min treadmill running was conducted in any phase, there were no effects on the general condition of the rats. Furthermore, there were no differences among the groups in terms of paw volume, but histologically, joint destruction was significantly suppressed in the PAL and T groups when compared with the control and PAS groups.

Based on these results, we concluded that treadmill running more effectively suppresses joint destruction during the established phase of RA than during the pre-arthritis phase.

We also analyzed the role of Cx43 as a key gene in synovial membrane inflammation in RA. Studies have shown that, in collagen-induced arthritis (CIA) rats, Cx43 induces joint destruction through pro-inflammatory cytokines such as TNF-α and IL-6, and that joint destruction is suppressed not only if Cx43 expression is suppressed in small interfering ribonucleic acid, but also by suppressing Cx43 expression through treadmill running as well [17,23,24]. Cx43 expression changes with mechanical stress such as stretching stimuli and shearing forces in cardiomyocytes and osteocytes [25,26]; therefore, it may also change due to the addition of mechanical stress to the synovial membranes by treadmill running. In this study, TNF-α expression in the synovial membranes of the T group was significantly lower than that of the control group, but there were no differences in TNF-α expression between the control and PAS groups. As with TNF-α, Cx43 expression in the synovial membranes of the T group was significantly lower than in the control and PAS groups.

Taking these results and the structural results into consideration together, we are of the opinion that treadmill running during the pre-arthritis phase is unable to suppress the sudden spike in Cx43 and TNF-α expression that has been described to occur in this phase, and does not completely suppress arthritis or joint destruction. However, treadmill running during the established phase can suppress the expression of these substances in their steady state, and thus suppress arthritis and joint destruction.

In this study, our analysis of the efficacy of treadmill running in suppressing bone destruction during each phase of RA found that with respect to bone morphometry, BV/TV, Tb.Th, and BMC/TV increased in the T group relative to the control group. MSV tended to be lower for the three groups that did treadmill running than for the control group, and it was especially low in the T group. These results made it clear that treadmill running was most effective in suppressing bone destruction in the established phase of RA. Furthermore, talus surface absorption area (Es/Rps)—which is an indicator of bone destruction—was suppressed in the T group. The T and PAL also groups had fewer cathepsin K immunostaining positive cells than the control and PAS groups.

Osteoclasts that differentiate from bone marrow-derived monocyte-macrophage progenitors play an important role in bone destruction in RA, and suppressing their differentiation is essential for suppressing bone destruction. Differentiation from progenitor cells into osteoclasts is facilitated by TNF-α [27], and because TNF-α is suppressed by mechanical stress [28], we conclude that in this study, bone destruction was suppressed through treadmill running during the established phase, in which mechanical stress exerted on the synovial membranes suppressed TNF-α expression, and the stress itself directly suppressed osteoclast differentiation. Furthermore, the receptor activator of the nuclear factor-kappa B ligand (RANKL) is also the key cytokine that induces osteoclast formation. In RA, osteoclasts are responsible for bone erosion, and they undergo differentiation and activation by RANKL, which is secreted by synovial fibroblasts, T cells, and B cells. Sato et al. reported that the inflammatory cytokines enhance RANKL expression in osteoclastogenesis-supporting cells and activate osteoclast precursor cells by synergizing with RANKL signaling [29]. Therefore, the decrease in the inflammatory cytokines, such as TNF-α, due to treadmill running may partly suppress osteoclast differentiation via RANKL.

The results of this study revealed the potential for efficacy of exercise therapy to differ depending on RA disease activity. We suggest that in phases with high disease activity, that is when arthritis is exacerbated, it may be possible to more effectively suppress arthritis and joint destruction by using pharmacotherapy in a central role, and at the same time using active exercise therapy during phases in which disease activity declines and inflammatory cytokine production is in a steady state.

There were several limitations to this study. First, we did not conduct a detailed study on the mechanisms through which treadmill running suppresses Cx43 and TNF-α. Second, when considering clinical applications, pharmacotherapy is necessary, but we did not study the efficacy of combined pharmacotherapy and exercise therapy. Third, we only investigated immunostaining with cathepsin K as an osteoclast marker without using tartrate-resistant acid phosphatase staining and measuring the serum C-telopeptide of type I collagen level.

## 4. Materials and Methods

This study was conducted in accordance with the animal research guidelines of the Kyoto Prefectural University of Medicine, Kyoto, Japan (code no. M29-22, 1 April 2017).

### 4.1. CIA Model

Rat CIA models have many points of similarity with human RA and are widely used for RA studies in vivo. To create such models for this study, type II collagen (CII; Collagen Research Center, Tokyo, Japan) and Freund’s incomplete adjuvant (FIA; Sigma-Aldrich, Saint Louis, MO, USA) were blended and emulsified on ice in a 1:1 ratio. Then, to create rat CIA models, 200 μL of the CII/FIA solution was intracutaneously injected into 32 8-week-old male Dark Agouti (DA) rats (Shimizu Laboratory Suppliers, Kyoto, Japan) at the base of the tail [30]. The rats weighed between 140 and 190 g.

### 4.2. Treadmill Running Protocol

Based on the study by Dekkers et al. [19], we divided all the induced rats into the following groups: the control group (*n* = 8), which was freely raised at random; and the pre-arthritis intervention short (PAS) (*n* = 8), pre-arthritis intervention long (PAL) (*n* = 8), and therapeutic intervention (T) (*n* = 8) groups, which were made to run on a treadmill (TMS8D; MEQUEST, Toyama, Japan) from day 14 to 28, day 14 to 42, and day 28 to 42, respectively. The running conditions for the PAS, PAL, and T groups were set for efficacy of suppression of joint destruction in rat OA models and rat RA models from a previous study: 5 times/week, speed of 12 m/min, 30 min/day [17,31]. Rats with CIA may have reduced activity due to inflammation and pain. To ensure uniform activity, we applied slightly electric stimulation to encourage the rat from behind if it stopped on the treadmill machine. In this study, all rats ran without electric stimulation, so they all had the same performance. The running protocols for each group are shown in Figure 6. The rats were raised in 12-h light:dark cycles and allowed free access to food and water. All the rats were sacrificed on day 42.

### 4.3. Body Weight, Paw Volume, and Clinical Score

Body weight, paw volume, and clinical score were measured on days 0, 3, 6, 9, and 12, and daily from then (Figure 2). Paw volume was measured with a water replacement plethysmometer (Unicom Japan, Tokyo, Japan). The clinical score was defined as follows: score 0, normal paw; score 1, inflammation and swelling of one toe; score 2, inflammation and swelling of > 1 toe with inflammation and swelling of the entire paw or mild swelling of the entire paw; score 3, inflammation and swelling of the entire paw; score 4, severe inflammation and swelling of the entire paw or ankylosed paw [32]. Specimens with maximum paw volume < 2.5 mL were excluded because immunization of those rats is insufficient. Thus, two rats were excluded, one in the control group and another in the T group.

### 4.4. Histochemical Analyses and Semi-Quantitative Analyses

After the end of the treadmill running protocol, the right ankle joints of the rats were removed and set in 4% paraformaldehyde (Wako, Osaka, Japan). They were then decalcified with 20% ethylenediaminetetraacetic acid and embedded in paraffin. The center of each ankle joint was sliced in 6-µm thick sagittal slices, and stained with H&E and safranin O. Arthritic changes, such as infiltration of inflammatory cells, synovial proliferation, destruction of articular cartilage, and bone erosion, were then evaluated histologically, and we measured the histological score as described by Weinberger et al. [33]. In brief, the infiltration of mononuclear cells (0–3 points) and histiocytes (0–3 points) into the synovium, cartilage destruction (0–3 points), and bone erosion (0–5 points) were measured.

### 4.5. Immunohistochemical Analyses

For immunohistochemistry of TNF-α, Cx43, and cathepsin K, paraffin-embedded joint tissue sections were de-paraffinized in xylene, rehydrated with graded alcohol, and immersed in 0.4 mg/mL of phosphate buffered saline (PBS). Antigen activation was conducted in 8-min intervals with proteinase K, and endogenous peroxidase activity block was conducted at 5-min intervals with 3% H_2_O_2_. Rabbit polyclonal anti-TNF-α antibodies (ab6671, Abcam, Cambridge, UK) at a concentration of 1:150, and anti-Cx43 antibodies (#3512, Cell Signaling Technology, Danvers, MA, USA) at a concentration of 1:50 were incubated overnight at 4 °C. Rabbit polyclonal anti-cathepsin K (111239-1-AP, Proteintech, Chicago, IL, USA) at a concentration of 1:2000 was incubated for 50 min at room temperature. The sections were incubated in Histofine Simple Stain Rat MAX-PO for 30 min at room temperature, after extensive washing with PBS. Immunostaining was detected by 3,3′-diaminobenzidine (DAB) staining. Counter staining was done with Mayer’s hematoxylin. We confirmed that there is no nonspecific staining. TNF-α and Cx43 staining areas of the synovial membrane interstitial cells were calculated using ImageJ^®^ with modifying a method reported by Mane et al. [34]. In detail, three fields of view with a magnification of 400× were taken randomly. The images were deconvoluted and showed only DAB immunoreaction. A standard threshold was maintained without any adjustment. The percentage of the immunostained-positive area was averaged. Validation of ImageJ^®^ analysis was performed by two expert orthopedists. Quantification of all the images was blinded.

### 4.6. Micro-Computed Tomography Analysis

The left ankle joints of the rats were fixed in 70% ethanol and scanned with a μ-CT system (TOSCANER-32300μFD, Toshiba, Tokyo, Japan). The reconstructed data sets were examined with three-dimensional data analysis software (TRI/3-D-BON, Ratoc System Engineering Co., Tokyo, Japan; *n* = 4 for each group) [35]. The regions of interest were defined in the whole talus bones. To analyze the talus, the following trabecular bone parameters in the whole talus were evaluated: bone volume fraction, Tb.Th, and bone mineral content per tissue volume (BMC/TV). We also assessed indirect parameters, including MSV, which is the mean volume of all the parts of an object that can be unobscured in all the directions from a point inside the object [36]. The degree of bone erosion and bone formation were also analyzed using 3D-μ-CT. Eroded bone surface per repaired bone surface (Es/Rps) on whole talus was determined with the software automatically according to the software program. We set the concave surface search range up to 0.5 mm, and the absorption surface extraction radius of curvature was 240 µm or less.

### 4.7. Statistical Analysis

All data were presented as the mean and standard deviation and analyzed with Statistical Package for Social Sciences (SPSS) 26.0 for Windows (SPSS Inc., Chicago, IL, USA). The data were analyzed by analysis of variance, and post hoc testing was performed with the Tukey–Kramer test. In all analyses, *p* < 0.05 was defined as statistically significant.

## 5. Conclusions

This study was the first to report on the varying effects of treadmill running, at different phases of RA, on the synovial membranes, articular cartilages, and bones of rat RA models. For RA patients as well, it may be possible for therapeutic exercise in the appropriate phases to increase bio-molecular suppression of arthritis through the suppression of inflammatory cytokine expression, and to consequently suppress joint destruction.

## Figures and Tables

**Figure 1 ijms-20-05100-f001:**
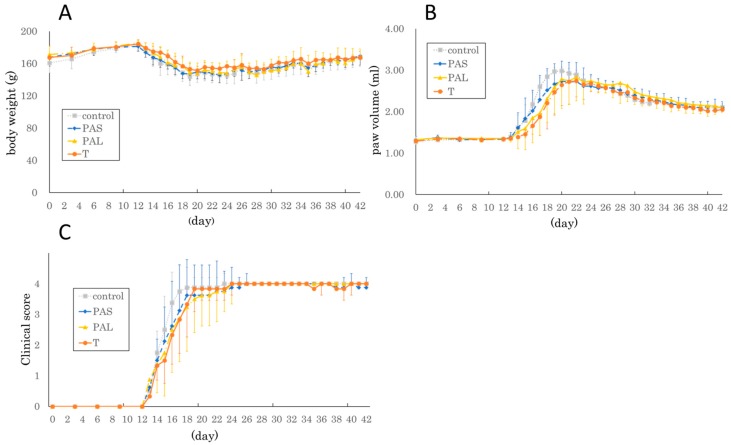
Kinetic change in (**A**) body weight, (**B**) paw volume, and (**C**) clinical score after immunization. The parameters were measured once every three days until day 12 and every day thereafter. There were no significant differences among the four groups (no intervention group, control; pre-arthritis intervention short group, PAS; pre-arthritis intervention long group, PAL; therapeutic intervention group, T) on all days.

**Figure 2 ijms-20-05100-f002:**
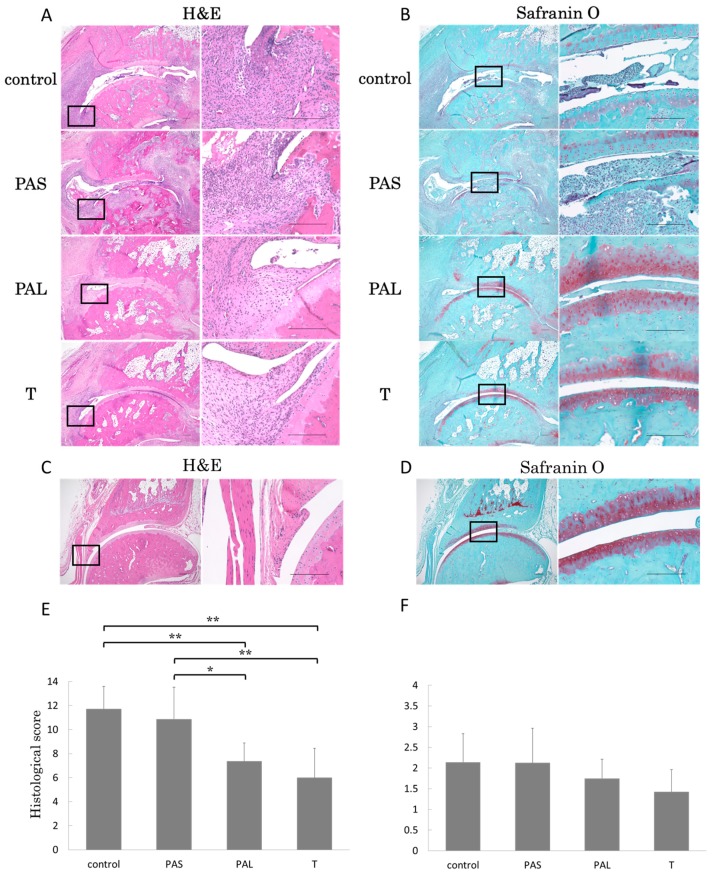
Representative micrographs of (**A**) hematoxylin and eosin, and (**B**) safranin O-stained sagittal sections. Representative micrographs of (**C**) hematoxylin and eosin, and (**D**) safranin O-stained sagittal sections in a normal rat without treadmill running. (**E**) The histological scores (mean ± standard deviation) and (**F**) only the cartilage evaluation scored based on the histological score (mean ± standard deviation) are shown. The PAL and T groups had suppressed destruction of the ankle joint more than the control and PAS groups. ** *p* < 0.01, * *p* < 0.05. Scale bar = 200 μm. The black spot represents the position in the high-magnification figure.

**Figure 3 ijms-20-05100-f003:**
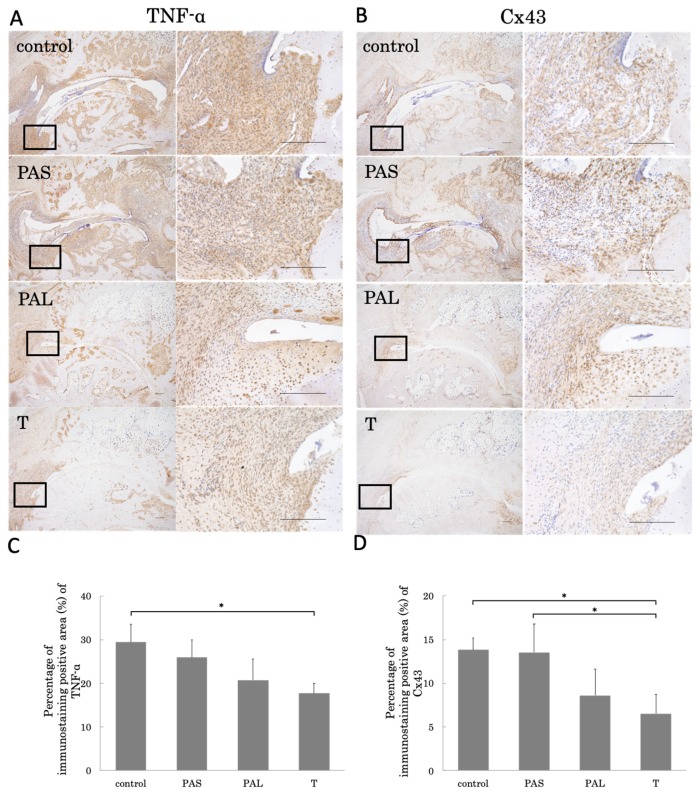
Representative micrographs of immunohistochemical staining for (**A**) TNF-α and (**B**) Cx43 are shown. All images were evaluated semi-quantitatively using ImageJ for (**C**) TNF-α and (**D**) Cx43. T group had significantly suppressed TNF-α and Cx43 expression. * *p* < 0.05. Scale bar = 200 μm. The black spot represents the position in the high-magnification figure.

**Figure 4 ijms-20-05100-f004:**
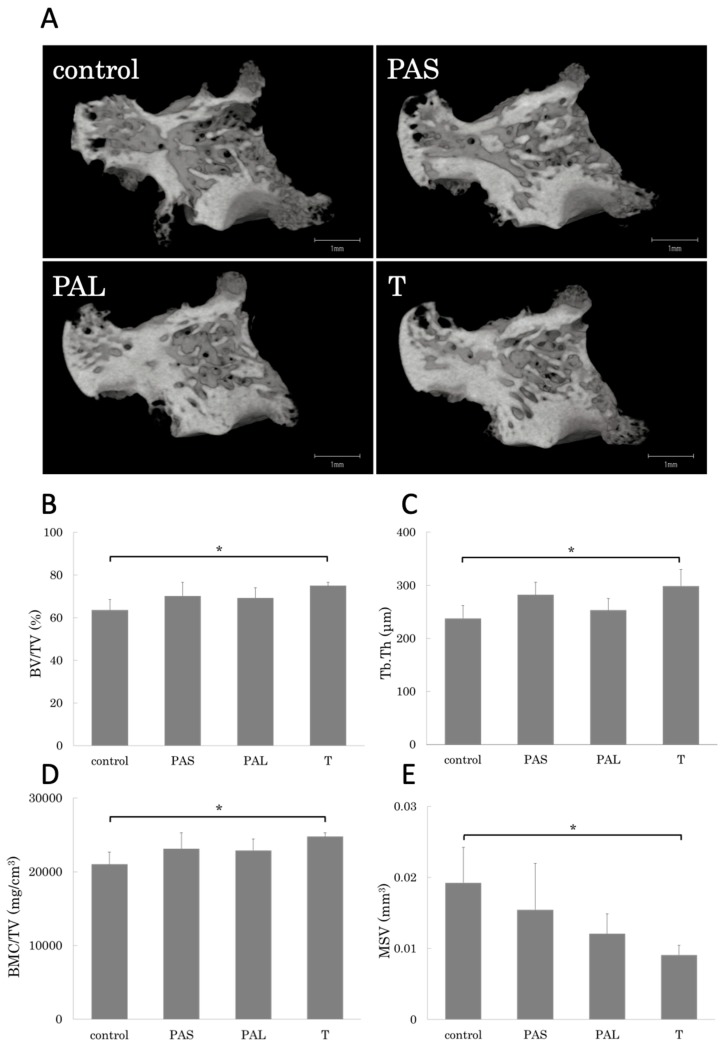
Representative three-dimensional reconstruction of **(A**) the sagittal sections of the talus architecture. Trabecular bone parameters such as (**B**) bone volume fraction (BV/TV), (**C**) trabecular thickness (Tb.Th), (**D**) bone mineral content per tissue volume (BMC/TV), and (**E**) marrow star volume (MSV) of the whole talus are shown. T group had improved bone loss. *n* = 4 in each group. * *p* < 0.05.

**Figure 5 ijms-20-05100-f005:**
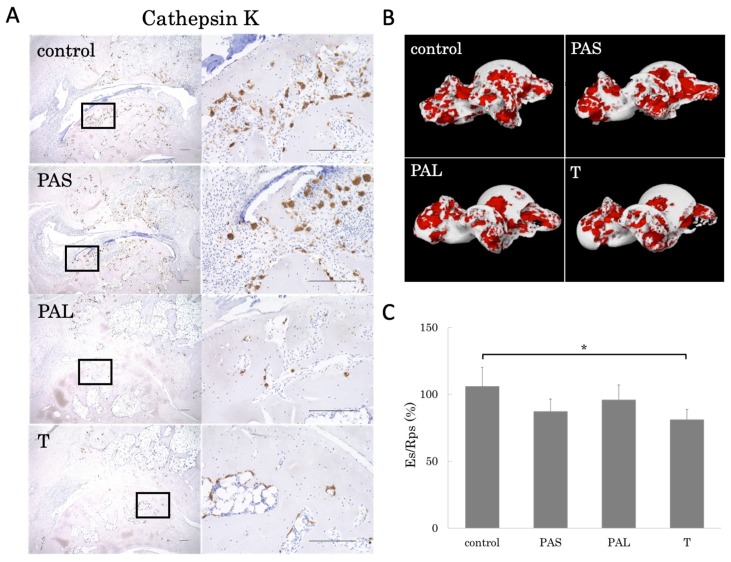
(**A**) Representative micrographs of cathepsin K immunohistochemical staining. Cathepsin K positive cells were fewer in the PAL and T groups. (**B**) Representative 3D reconstruction of bone erosion area in the whole talus architecture. Red area is bone erosion area. (**C**) The eroded bone surface (Es) and repaired bone surface (Rps) was calculated using 3D-μ-CT, and Es/Rps values are shown for the four groups. Es/Rps was significantly lower in the T group compared to the control group. * *p* < 0.05. Scale bar = 200 μm. The black spot represents the position in the high-magnification figure.

**Figure 6 ijms-20-05100-f006:**
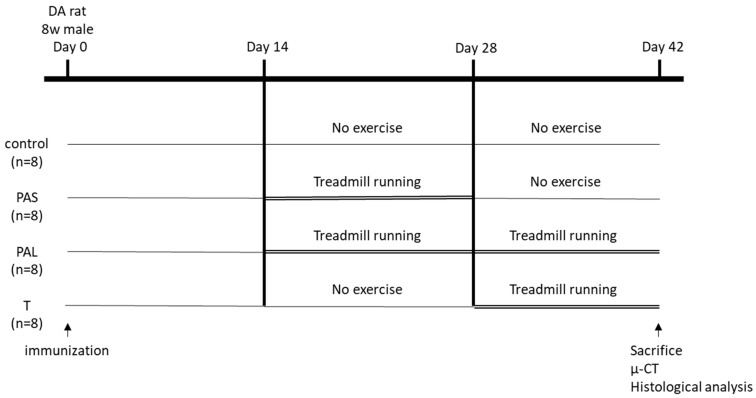
Experimental protocols. Eight-week-old male Dark Agouti rats were randomly divided into four groups: control, PAS, PAL, and T groups. PAS (*n* = 8), run from day 14 to 28; PAL (*n* = 8), run from day 14 to 42; and T (*n* = 8), run from day 28 to 42.

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
