# Peer review of "Treadmill Running in Established Phase Arthritis Inhibits Joint Destruction in Rat Rheumatoid Arthritis Models"

_ijms, 2019, doi:10.3390/ijms20205100_

Round 1
Reviewer 1 Report
This is an interesting paper and the main finding may be of general interest, although it closely mirrors findings they have previously published (Int. J. Mol. Sci. 2018, 19(6), 1653; https://doi.org/10.3390/ijms19061653). However, whilst parts of the data are sound there are a number of flaws in the presentation, validation and analysis of the data presented. One key area that should be considered in the impact of pain on exercise. Do the CIA mice show equal activity on the treadmill and how is this assessed. Mice with worse inflammation and pain may show reduced exercise behaviour. If this was recorded and normalised per mouse this should be clarified and reported. Secondly, analysis of bone parameter secondary to micro-ct changes are limited. As far as this reviewer is aware, Cathepsin K, whilst a marker of osteoclast maturation is not usually the primary way of looking at osteoclasts at sites of inflammation. TRAP staining on the histology would address this and allow some simple assessment of OC numbers relative to bone surface area if OC are believed to be the main driver of phenotype difference in bone. Otherwise, there are some important points that the authors need to address and correct around this data prior to publication.
Major:
Figure 1. Group abbreviations are missing from the results discussion and legend reducing clarity of results.
Methods: It is clear to me that all groups were CIA with different therapeutic exercise interventions but this should be made clearer in methods and legends. Rather than calling the control group CIA, you should call this something like no intervention control as all animals are CIA. This would help clarity and interpretation.
Fig 1: please include a clinical score of joint inflammation, usually considering changes in rat behaviour, mobility, weight and signs of pain behaviour. It is difficult to assess how rats respond to exercise without having a measure of inflammation and pain beyond swelling
Fig 2: details of histology scoring need to be explained and included in brief rather than reference alone. It is currently unclear what these scores really mean making interpretation impossible?
Fig 2-5: please include a non inflamed control for comparison and include either additional higher magnification images or increase magnification for all joints to regions of interest. At the moment it is not always clear what to focus on in the histology images. Better magnification and arrows to highlight regions of interest would help.
Fig 2: Please separate scores for cartilage versus histology scoring in H&E sections. There is clearly a cartilage phenotype that is poorly reported by the scoring
Fig 3: figures are poorly labelled and bad resolution. No labels to indicate which stain you have performed. This is missing IgG controls for non specific staining. Again higher magnification images and arrows to demonstrate staining would help for clarity.
Fig 3: legend and methods for how image J analysis was performed is absent. More details on criteria, controls and validation of image J staining are needed prior to publication
Fig 4: can you add in numbers of mice used for powering micro-Ct in legend. Please also add images of representative trabecular bone reconstructions for consideration
Fig 5: no IgG controls for Kathepsin K. Where is the literature on efficacy for this as a measure of resorption. Better to measure CTX-1 in serum or perform TRAP staining and assess OC numbers relative to bone surface area at the joint. Either talk down significance of this or perform better measures of bone resorption. In particular, these data should focus on osteoclasts at sites of bone erosion in this model. In particular, good magnification and analysis of appropriate sites of inflammation with arrows are missing.
Fig 5: Validation of figures b and c is missing. The methodology needs to be much clearer. How exactly was this done? Is this a well published and validated method of assessing bone erosions? Something better to justify this is required
Author Response
Response to Reviewer 1 Comments
We appreciate your review comments. Point-by-point replies to the comments are listed in this letter.
Major:
Figure 1. Group abbreviations are missing from the results discussion and legend reducing clarity of results.
〈Reply〉
Thank you for pointing out. We added group abbreviations in the results discussion and figure legend as follow.
Line 87-92 in the revised manuscript
Body weight in each group (no intervention [control] group, pre-arthritis intervention short [PAS] group, pre-arthritis intervention long [PAL] group, and therapeutic intervention [T] group) gradually increased from day 0 to day 12, decreased from day 12 to day 20, and increased again starting on day 21. Paw volume in each group increased from day 14, reaching its maximum between day 20 and day 23, and gradually decreasing thereafter. No significant differences in either body weight or paw volume were observed among groups from day 0 to day 42 (Figure 1A-C).
Line 96-99 in the revised manuscript
There were no significant differences among the 4 groups (no intervention group, control; pre-arthritis intervention short group, PAS; pre-arthritis intervention long group, PAL; therapeutic intervention group, T) on all days.
Methods: It is clear to me that all groups were CIA with different therapeutic exercise interventions but this should be made clearer in methods and legends. Rather than calling the control group CIA, you should call this something like no intervention control as all animals are CIA. This would help clarity and interpretation.
〈Reply〉
Thank you very much. As you indicated, the “collagen-induced arthritis (CIA) group” was changed to the “no intervention (control) group”.
Line 87-92 in the revised manuscript
Body weight in each group (no intervention [control] group, pre-arthritis intervention short [PAS] group, pre-arthritis intervention long [PAL] group, and therapeutic intervention [T] group) gradually increased from day 0 to day 12, decreased from day 12 to day 20, and increased again starting on day 21. Paw volume in each group increased from day 14, reaching its maximum between day 20 and day 23, and gradually decreasing thereafter. No significant differences in either body weight or paw volume were observed among groups from day 0 to day 42 (Figure 1A-C).
Line 96-99 in the revised manuscript
There were no significant differences among the 4 groups (no intervention group, control; pre-arthritis intervention short group, PAS; pre-arthritis intervention long group, PAL; therapeutic intervention group, T) on all days.
Line 103, 106, 107, 108, 116, 120, 121, 130, 132, 141, 144, 145, 152, 156, 164, 175, 176, 177, 187, 188, 192, 230, 244, and 254 in the revised manuscript
We changed “CIA” to “control”.
Fig 1: please include a clinical score of joint inflammation, usually considering changes in rat behaviour, mobility, weight and signs of pain behaviour. It is difficult to assess how rats respond to exercise without having a measure of inflammation and pain beyond swelling
〈Reply〉
We added a clinical score according to your proposal. As you indicated, joint inflammation may change rat behaviour, mobility, weight and signs of pain behavior. Therefore, we applied slightly electric stimulation to encourage the rat from behind if it stopped on the treadmill machine. In this study, all rats ran without electric stimulation, so they all had the same performance. We added the sentences as below.
Line 93-94 in the revised manuscript
We added a graph of clinical score as Figure 1C.
Line 235-238 in the revised manuscript
Rats with CIA may have reduced activity due to inflammation and pain. To ensure uniform activity, we applied slightly electric stimulation to encourage the rat from behind if it stopped on the treadmill machine. In this study, all rats ran without electric stimulation, so they all had the same performance.
Line 249-252 in the revised manuscript
The clinical score was defined as follows: score 0, normal paw; score 1, inflammation and swelling of one toe; score 2, inflammation and swelling of >1 toe with inflammation and swelling of the entire paw or mild swelling of the entire paw; score 3, inflammation and swelling of the entire paw; score 4, severe inflammation and swelling of the entire paw or ankylosed paw [32].
Fig 2: details of histology scoring need to be explained and included in brief rather than reference alone. It is currently unclear what these scores really mean making interpretation impossible?
〈Reply〉
I'm sorry, the reference was wrong. I corrected the references and added measuring method of histological score in briefly.
Line 259-264 in the revised manuscript
Arthritic changes, such as infiltration of inflammatory cells, synovial proliferation, destruction of articular cartilage, and bone erosion, were then evaluated histologically, and we measured the histological score as described by Weinberger et al [33]. In brief, the infiltration of mononuclear cells (0-3 points) and histiocytes (0-3 points) into the synovium, cartilage destruction (0-3 points), and bone erosion (0-5 points) were measured.
Line 411-413 in the revised manuscript
33. Weinberger, A.; Halpern, M.; Zahalka, M.A.; Quintana, F.; Traub, L.; Moroz, C. Placental immunomodulator ferritin, a novel immunoregulator, suppresses experimental arthritis. Arthritis Rheum. 2003, 48, 846–853.
Fig 2-5: please include a non inflamed control for comparison and include either additional higher magnification images or increase magnification for all joints to regions of interest. At the moment it is not always clear what to focus on in the histology images. Better magnification and arrows to highlight regions of interest would help.
〈Reply〉
We added the images of normal rat in figure 2. We highlighted the region of interest with a rectangle, and added high-magnification images in figure 2-5.
Fig 2: Please separate scores for cartilage versus histology scoring in H&E sections. There is clearly a cartilage phenotype that is poorly reported by the scoring.
〈Reply〉
We separated scores for cartilage, and added figure 2(F). Figure 2E represents the histological scores and figure 2F represents only the cartilage evaluation scored based on the histological score.
Line 113-115 in the revised manuscript
E) The histological scores (mean ± standard deviation) and F) only the cartilage evaluation scored based on the histological score (mean ± standard deviation) are shown.
Fig 3: figures are poorly labelled and bad resolution. No labels to indicate which stain you have performed. This is missing IgG controls for non specific staining. Again higher magnification images and arrows to demonstrate staining would help for clarity.
〈Reply〉
We added labels to the figure, and increased resolution. We preliminarily confirmed that there is no non specific staining. The region of interest was highlighted with a rectangle, and we added high-magnification images.
Line 276-277 in the revised manuscript
We confirmed that there is no non specific staining.
Fig 3: legend and methods for how image J analysis was performed is absent. More details on criteria, controls and validation of image J staining are needed prior to publication
〈Reply〉
We added the details of the evaluation method using Image J into materials and methods.
Line 277-283 in the revised manuscript
TNF-α and Cx43 staining areas of the synovial membrane interstitial cells were calculated using ImageJ® with modifying a method reported by Mane et al [34]. In detail, three fields of view with a magnification of 400× were taken randomly. The images were deconvoluted and showed only DAB immunoreaction. A standard threshold was maintained without any adjustment. The percentage of the immunostained-positive area was averaged. Validation of ImageJ® analysis was performed by two expert orthopaedists. Quantification of all the images was blinded.
Line 414-416 in the revised manuscript
34. Mane, D.R.; Kale, A.K.; Belaldavar, C. Validation of immunoexpression of tenascin-C in oral precancerous and cancerous tissues using ImageJ analysis with novel immunohistochemistry profiler plugin: An immunohistochemical quantitative analysis. J. Oral Maxillofac. Pathol. 2017, 21, 211-217.
Fig 4: can you add in numbers of mice used for powering micro-Ct in legend. Please also add images of representative trabecular bone reconstructions for consideration
〈Reply〉
We added numbers of rats used for micro-CT to “materials and methods” and “legend”. In addition, images of representative trabecular bone reconstructions were added to figure 4 as Figure 4A.
Line 285-288 in the revised manuscript
The left ankle joints of the rats were fixed in 70% ethanol and scanned with a μ-CT system (TOSCANER-32300μFD, Toshiba, Tokyo, Japan). The reconstructed data sets were examined with three-dimensional data analysis software (TRI/3-D-BON, Ratoc System Engineering Co., Tokyo, Japan; n = 4 for each group) [35].
Fig 5: no IgG controls for Kathepsin K. Where is the literature on efficacy for this as a measure of resorption. Better to measure CTX-1 in serum or perform TRAP staining and assess OC numbers relative to bone surface area at the joint. Either talk down significance of this or perform better measures of bone resorption. In particular, these data should focus on osteoclasts at sites of bone erosion in this model. In particular, good magnification and analysis of appropriate sites of inflammation with arrows are missing.
〈Reply〉
We confirmed that there is no non specific staining. We added the literature on efficacy for cathepsin K in line143, and weakened expression. The region of interest was placed at osteoclasts at sites of bone erosion. We highlighted this region with a rectangle, and added a high-magnification image. We added the limitation that we only investigated immunostaining with cathepsin K as one of the osteoclast marker without using TRAP staining and measuring the serum CTX-1 level.
Line 154-155 in the revised manuscript
To assess its effects on bone metabolism, we conducted immunostaining with cathepsin K, one of the osteoclast marker that serves as an indicator of the degree of bone resorption [21, 22].
Line 215-217 in the revised manuscript
Third, we only investigated immunostaining with cathepsin K as an osteoclast marker without using tartrate-resistant acid phosphatase staining and measuring the serum C-telopeptide of type I collagen level.
Line 379-382 in the revised manuscript
21. Drake, M.T.; Clarke, B.L.; Ourler, M.J.; Khosla, S. Cathepsin K inhibitors for osteoporosis: biology, potential clinical utility, and lessons learned. Endcr. Rev. 2017, 38, 325–350.
22. Wilson, S.R.; Peters, C.; Saftig, P.; Bromme, D.; Cathepsin K activity-dependent regulation of osteoclast actin ring formation and bone resorption. J. Biol. Chem. 2009, 284, 2584-2592.
Fig 5: Validation of figures b and c is missing. The methodology needs to be much clearer. How exactly was this done? Is this a well published and validated method of assessing bone erosions? Something better to justify this is required.
〈Reply〉
Thank you for pointing out. We added the citation and details of methodology of Figure 5b, c. We used TRI / 3D-BON software because that software is widely used and published for bone morphometry.
Line 286-288 in the revised manuscript
The reconstructed data sets were examined with three-dimensional data analysis software (TRI/3-D-BON, Ratoc System Engineering Co., Tokyo, Japan; n = 4 for each group) [35].
Line 293-296 in the revised manuscript
Eroded bone surface per repaired bone surface (Es/Rps) on whole talus was determined with the software automatically according to the software program. We set the concave surface search range up to 0.5 mm, and the absorption surface extraction radius of curvature was 240 µm or less.
Line 417-419 in the revised manuscript
35. Munemoto, M.; Kido, A.; Sakamoto, Y.; Inoue, K.; Yokoi, K.; Shinohara, Y.; Tanaka, Y. Analysis of trabecular bone microstructure in osteoporotic femoral heads in human patients: in vivo study using multidetector row computed tomography. BMC Musculoskelet Disord. 2016, 17, 13.
Reviewer 2 Report
The present manuscript under review titled; “Treadmill Running in Established Phase Arthritis Inhibits Joint Destruction in Rat Rheumatoid Arthritis Models” The data suggest that exercise therapy can be considered as a potential therapeutic candidate in inhibiting arthritis and joint destruction in rheumatoid arthritis at established phase.In general data is appropriately interpreted and analyzed. However, there are several instances where additional data would be necessary to support the hypothesis and elevate the impact of the study.
Discussion: The authors concluded that “bone destruction was suppressed through treadmill running during the established phase, which mechanical stress exerted on the synovial membranes suppressed TNF-α expression, and the stress itself directly suppressed osteoclast differentiation”. However, receptor activator of nuclear factor-kappa B ligand (RANKL) is also the key cytokine that induces osteoclast formation. In RA, osteoclasts are known to be responsible for bone erosion and undergo differentiation and activation by RANKL, which is secreted by synovial fibroblasts, T cells, and B cells. The authors should demonstrate that effect of exercise therapy on the RANKL expression in RA model.
Author Response
Response to Reviewer 2 Comments
We appreciate your review comments. Point-by-point replies to the comments are listed in this letter.
Discussion: The authors concluded that “bone destruction was suppressed through treadmill running during the established phase, which mechanical stress exerted on the synovial membranes suppressed TNF-α expression, and the stress itself directly suppressed osteoclast differentiation”. However, receptor activator of nuclear factor-kappa B ligand (RANKL) is also the key cytokine that induces osteoclast formation. In RA, osteoclasts are known to be responsible for bone erosion and undergo differentiation and activation by RANKL, which is secreted by synovial fibroblasts, T cells, and B cells. The authors should demonstrate that effect of exercise therapy on the RANKL expression in RA model.
〈Reply〉
Thank you for your review. As you pointed out, RANKL is also an important cytokine for osteoclast differentiation. We added sentences and a reference about RANKL.
Line 199-206 in the revised manuscript
Furthermore, the receptor activator of the nuclear factor-kappa B ligand (RANKL) is also the key cytokine that induces osteoclast formation. In RA, osteoclasts are responsible for bone erosion, and they undergo differentiation and activation by RANKL, which is secreted by synovial fibroblasts, T cells, and B cells. Sato et al reported that the inflammatory cytokines enhance RANKL expression in osteoclastogenesis-supporting cells and activate osteoclast precursor cells by synergizing with RANKL signaling [29]. Therefore, the decrease in the inflammatory cytokines, such as TNF-α, due to treadmill running may partly suppress osteoclast differentiation via RANKL.
Line 399-401 in the revised manuscript
29. Sato, K.; Suematsu, A.; Okamoto, K.; Yamaguchi, A.; Morishita, Y.; Kadono, Y.; Tanaka, S.; Kodama, T.; Akira, S.; Iwakura, Y.; Cua, D.J.; Takayanagi, H. Th17 functions as an osteoclastogenic helper T cell subset that links T cell activation and bone destruction. J. Exp. Med. 2006, 203, 2673-2682.
Round 2
Reviewer 1 Report
The authors have adequately addressed all concerns and comments raised with the manuscript by this reviewer and I am happy to recommend for publication
Reviewer 2 Report
The authors of the manuscript have adequately answered to all the raised questions. The new additions and modifications have significantly improved the quality of the manuscript.